# Evaluation of a Pilot College Student-Based Physical Activity Implementation Strategy in a Rural Middle School

**DOI:** 10.3390/ijerph21121645

**Published:** 2024-12-10

**Authors:** Megan M. Kwaiser, Andrew M. Medellin, Janette M. Watkins, Janelle M. Goss, James M. Hobson, Vanessa M. Martinez Kercher, Kyle A. Kercher

**Affiliations:** 1Department of Kinesiology, School of Public Health-Bloomington, Indiana University, Bloomington, IN 47405, USA; mkwaiser@iu.edu (M.M.K.); janhynes@iu.edu (J.M.W.); jangoss@iu.edu (J.M.G.); 2Department of Epidemiology & Biostatistics, School of Public Health-Bloomington, Indiana University, Bloomington, IN 47405, USA; andrewmmedellin@gmail.com; 3Program in Neuroscience, College of Arts and Sciences, Indiana University, Bloomington, IN 47405, USA; 4White River Valley Middle School, Lyons, IN 47433, USA; jmhobson91@gmail.com; 5Department of Health & Wellness Design, School of Public Health-Bloomington, Indiana University, Bloomington, IN 47405, USA; vkercher@iu.edu

**Keywords:** implementation science, feasibility testing, sport-based youth development, public health

## Abstract

Background: Physical inactivity in the U.S. poses a significant risk of developing chronic health factors associated with cardiovascular disease. Children from rural communities are especially vulnerable to inactivity. The Hoosier Sport program aims to address this by working to increase physical activity in 6th and 7th grade students in a rural Indiana middle school. Hoosier Sport uses sport participation coupled with health education delivered by college-service learning students to establish healthy behaviors that children can sustain throughout their life. The purpose of this prospective longitudinal study was to evaluate the implementation of Hoosier Sport in a rural middle school, using a multi-component evaluation approach. Methods: This prospective program evaluation study utilized The Consolidated Framework for Implementation Research (CFIR) to assess feasibility outcomes such as recruitment, retention, fidelity, attendance, acceptability, and cost. CFIR was incorporated through surveys completed by Hoosier Sport team members to identify facilitators and barriers. Fidelity was measured using SOSPAN and SOFIT tools. SOSPAN (System for Observation of Staff Promotion of Activity and Nutrition) monitored staff interactions with children during physical education classes. SOFIT (System of Observing Fitness Instruction Time) evaluated the duration and type of activities in each lesson context. For our descriptive analysis, we calculated means and standard deviation for continuous variables and percentages for categorical variables. Results: All feasibility measures met or exceeded the a priori threshold, indicating high success. Fidelity was high among college student implementers and child participants. SOSPAN showed that staff did not use physical activity as punishment, engaged in physical activity 62.5% of the time, provided verbal encouragement 87.5% of the time, and used elimination games only 2.5% of the time. SOFIT revealed significant promotion of moderate-to-vigorous physical activity, with 94% during the 4-week strength training intervention and 95% during the 4-week basketball intervention. The barrier buster tool identified general agreement with most statements, indicating promising system-level acceptability. Conclusion: The study results demonstrate successful feasibility, high fidelity, and promising system-level acceptability. These findings underscore the importance of continued refinement and repeated evaluation of the program in alignment with the ORBIT model. The use of college student implementers presents a sustainable model that benefits all participants involved.

## 1. Contributions to the Literature

The present study explored the use of a self-determination theory (SDT) framework to implement a multicomponent intervention. SDT stands as one of the most prevalent contemporary social cognitive motivational theories in physical activity.

Leveraging existing school assets and school-specific implementation strategies may be most beneficial for physical activity promotion. The present study aimed to enhance PE through service-learning students with a community co-design curriculum.Stakeholder engagement was suggested as a key facilitator for policy changes at the school level. The present study explored the implementation of a program co-designed by key stakeholders and reported on policy system environment data.

## 2. Introduction

Physical inactivity is a major public health issue that is becoming increasingly prevalent in children worldwide [1]. In the United States, less than 25% of children between the ages of 6 and 17 meet the recommended 60 min of physical activity every day [2]. Physical inactivity is one of the primary modifiable behaviors for the prevention of cardiovascular disease (CVD) [3,4]. Importantly, physical inactivity also increases the likelihood of developing other chronic health issues, such as diabetes and some forms of cancer, and impacts quality of life [5]. Furthermore, physical inactivity has also been linked to decreased cognitive function and increased mental health issues among youth [6].

Schools can be effective environments for efforts designed to help children achieve the recommended amount of moderate-to-vigorous physical activity (MVPA) as they provide support and structure [7]. This is especially true for schools located in rural settings where opportunities for structured physical activity outside of school are often limited, based on lack of access to facilities, trained coaches/professionals, sidewalks, transportation, and limited economic resources [8]. Schools have relatively high attendance, which overcomes many of the transportation and socioeconomic challenges associated with trying to transport rural children to non-school-based physical activity and sport programs [7]. Structured settings, such as schools, have been identified in a systematic review to be important contributors to total physical activity, MVPA, and public health [7]. Further, regular physical education classes have been found to significantly contribute to the recommended levels of MVPA [9].

Children receive a substantial amount of physical activity and associated health benefits while participating in sport programs [10]. Sport-based youth development programs use sports to “facilitate learning and life skill development in youth” [11] and have been found to have a positive impact on physical activity levels, socialization, emotions, physical well-being, cognition, and mental health [12,13,14,15,16]. However, the benefits of youth sports are not accessible to all populations of children, specifically children in rural and low-socioeconomic communities [8,17]. This makes sport-based youth development programs particularly important for increasing accessibility to potentially beneficial programs. One such program for youth from under-resourced rural backgrounds, called Hoosier Sport, was evaluated within the current study. Hoosier Sport is a sport-based youth development program that uses sports to engage youth in physical activity-related programs. Unlike many other physical activity programs, Hoosier Sport uniquely integrates leadership skill development and health education into its curriculum, ensuring a holistic approach to youth development. Additionally, Hoosier Sport is a campus–community partnership between a large Midwestern university and an under-resourced rural school district, with a service-learning program that ensures long-term sustainability. This sport-based program is in the early stages of implementation feasibility testing within the Obesity-Related Behavioral Intervention Trials (ORBIT) model [18].

Trained college student implementers present an innovative and potentially impactful approach to delivering sport-based youth development programming to children. This initiative presents an unusual opportunity for universities which frequently lack community-based programs and supports cost effectiveness, fidelity, and sustainability. College student implementers provide a community-driven opportunity that benefits the college students, children, and community alike. This format enables college students to gain valuable experience through community partnerships, offers the children college student role models through modeling and supportive behaviors, and enhances the overall quality of sport-based youth development in the community. College students benefit from community partnerships by gaining experience that future employers and graduate programs are looking for, while the children benefit from the well-recognized impact of college student role models and supporters [19]. Children often see young adults as more credible and relatable and are more likely to adopt their behaviors and messages about physical activity [20,21]. Communities and college students willing to work together for the common good of children makes this a mutually beneficial approach.

The Consolidated Framework for Implementation Research (CFIR) was used to guide the present study. CFIR is an implementation framework with five domains that are used in various combinations for the assessment of implementation strategies in health research (i.e., intervention characteristics, inner setting, outer setting, individuals, and process domains) [22]. Most of the domains took an implementer-focused approach, as child-facing clinical outcomes are published elsewhere [23]. Intervention characteristics were assessed, including feasibility, cost, and the implementers’ perceptions of barriers to and facilitators of implementation. The inner setting domain was used to inform recruitment strategies and assess retention through the pre- and post-surveys. The outer setting was assessed during the previously published physical activity-based needs assessment in the target community [24]. The individual characteristics were assessed through systematic observations of the intervention delivery (methods described later). The process domain guided assessment of fidelity throughout this study.

The purpose of this prospective longitudinal study was to evaluate the implementation of Hoosier Sport in a rural middle school using a multi-component evaluation approach. We collected implementation data on Hoosier Sport before, during, and after implementation of an 8-week sports-based youth development program that included two different sports sections (strength training and basketball). The primary aim was to examine the feasibility of Hoosier Sport (i.e., multiple trial- and intervention-related feasibility indicators) for delivering physical activity programming to children. Our secondary aim was to identify facilitators and barriers to implementing Hoosier Sport in the community, leading to constructive solutions that leverage facilitators and address barriers. From an exploratory standpoint, this study aimed to evaluate the fidelity of physical activity delivery during the program classes. This study addressed existing knowledge gaps by exploring the implementation of sport-based programming in under-resourced rural settings, a context that has been under-represented in the literature. By identifying practical implementation challenges and opportunities, this study contributed to the development of tailored, scalable solutions for delivering health-promoting physical activity interventions in similar populations. Our overarching hypothesis was that the data collected would show that Hoosier Sport implementation was feasible, delivered as intended, and would identify refinement opportunities for future intervention testing.

## 3. Methods

### 3.1. Conceptual Framework

The present evaluation study was guided by the self-determination theory (SDT). The mini theory of psychological needs satisfaction within the self-determination theory identifies three basic psychological needs (autonomy, competence, and relatedness) that promote well-being [25]. Autonomy is the need of an individual to feel in control of the choices for their behaviors and goals. Competence is the need for an individual to feel effective and capable in their actions and abilities. Relatedness is the need for an individual to have a sense of belonging or to feel connected with others. When all three of these psychological needs are met during physical activity programs, it increases intrinsic motivation in physical activity [26]. Intrinsic motivation among children in physical activity is a precursor for continued participation of physical activity and the development of the necessary skills and knowledge to continue to practice healthy habits in adulthood [27]. Figure 1 provides more detail on the conceptual model.

### 3.2. Setting and Sample

To determine the sample size of the pilot study, a power analysis was conducted in G*Power with a significance of 0.05 and a targeted power of 0.7. The analysis indicated that at least 30 participants were needed. This evaluation study included middle school children (*n* = 33) from a rural Midwestern middle school as participants in the intervention and Hoosier Sport research team members as implementers of the intervention (*n* = 7). The study participants were 6th and 7th graders who participated in the program during their physical education classes, representing a manageable sample size suitable for assessing feasibility and gathering preliminary data in a pilot study. Sixth and seventh graders who were enrolled in PE that semester were recruited at their middle school through the distribution of flyers and discussions with Hoosier Sport team members during lunch periods and class time. Guardians were contacted for approval for their children to participate in Hoosier Sport. Children who did not complete a Physical Activity Readiness Questionnaire (PAR-Q) or receive parental consent were not allowed to participate. Demographic data collected from child participants included age, gender, and race/ethnicity, providing insights into the diversity of the sample and ensuring alignment with the program’s goal of inclusivity in rural, under-resourced settings. The program’s evaluation also took place at the middle school. Hoosier Sport research team members included undergraduate and graduate students as well as faculty and staff from Indiana University, allowing for consistent training, close collaboration, and effective delivery of the intervention. This study was conducted in accordance with the Declaration of Helsinki, with ethical approval obtained from the Institutional Review Board at Indiana University (#18784).

### 3.3. Procedures

#### 3.3.1. Child Participants

Child participants were given consent from parents before having information collected and assenting to participate in data collection. This document included the purpose of the study, requirements of the study, and the potential risks and benefits from participating in the study. Information collected from the participants included demographic information and an adapted questionnaire. Children participated twice a week in a 45 min enhanced physical education class, where they developed and practiced skills in specific sports, such as strength training or basketball. Children’s type of physical activity was assessed during class to determine the amount of time spent in each lesson context (i.e., games, fitness, management) and the type of activities observed within each lesson context (i.e., standing, walking/moderate, walking/vigorous, vigorous).

The evaluations and questionnaires were conducted under a structured schedule designed to align with the Hoosier Sport program’s timeline and activities. SOSPAN scans were performed every third class session to assess the fidelity of physical activity delivery throughout the 8-week program. Child participants completed data collection at two time points, occurring one week before the program started and one week after its conclusion, respectively: baseline (Week 0) and post-program (Week 9). Implementers also completed surveys about their service-learning experience and beliefs at these same time points (Week 0 and Week 9) to capture changes in their perspectives and evaluate their role in the program.

Training for implementers and interviews with clinicians was conducted four weeks prior to the program, ensuring they were adequately prepared to deliver the intervention. The Hoosier Sport program itself was held twice weekly, on Tuesdays and Thursdays, during the students’ regularly scheduled physical education classes. All evaluations and data collection procedures were conducted on-site at the middle school, providing a familiar and convenient environment for participants. Child assessments and implementer surveys were completed individually to ensure privacy and accuracy, while SOSPAN observations occurred in the group class setting to capture real-time interactions and program dynamics. This schedule and structure facilitated a comprehensive evaluation of the program’s implementation and impact.

#### 3.3.2. Implementers

Hoosier Sport team members were recruited using in-class presentations, flyers, and word of mouth. Team members watched training videos to learn appropriate skill development, appropriate coaching cues for children, and lesson plan implementation for that semester’s sports. After each video, there was a short knowledge assessment conducted through Qualtrics to evaluate comprehension and reinforce key concepts. Team members also participated in a team training event at the beginning of the school year to cover the mission, values, emergency procedures, teaching fundamentals, positive youth development principles, and data collection techniques. These comprehensive training components were designed to enhance team members’ confidence and capability in implementing the program effectively. All training materials were developed by three certified coaches with expertise and experience in youth sport coaching. Hoosier Sport team members completed an adapted version of the barrier buster tool (described below) to identify their perceptions of barriers and facilitators of program implementation. This information was collected pre-implementation of the Hoosier Sport program and mid-implementation. The barrier buster tool was not collected post-implementation because the time frame between an additional time point was too short to identify meaningful changes that could occur within the intervention. Participants of the questionnaire included team members who directly facilitated program application and those who did not.

**Fidelity-related Procedures.** Team members who directly facilitated program application were observed using the System for Observation of Staff Promotion of Activity and Nutrition (SOSPAN) to assess the fidelity of physical activity delivery in program classes. Observers using the SOSPAN tool underwent comprehensive training, which included education on the background, purpose, and proper application of the tool. This training included practice sessions in both classroom and public settings to ensure familiarity and accuracy. Trained observers conducted 3 total scans per new activity (e.g., games, drills) per observation day, focusing on staff behaviors, management, and child participation. Observers followed classes throughout their time in the Hoosier Sport program, conducting scans during each scheduled activity on four separate days during the 8-week program. Observation days were not back to back and began at the same time as the organized program and ended with the conclusion of class. The SOSPAN passive observational tool was adapted for use in the Hoosier Sport program as the tool is primarily used for after school programs (ASPs). This adaptation ensured the tool’s relevance and applicability in capturing fidelity data within the distinct context of a rural middle school setting. Two observers conducted scans at the same time, one being the main observer and the other being a reliability scanner. The reliability scanner was used to ensure objectivity and try to eliminate subjective bias from the main observer. The scans were compared at the end of each scanning session to determine if >80% of the items found in the scans were agreed upon and matching in the checkboxes. The established validity of the SOSPAN tool, combined with rigorous training and reliability checks, ensured robust and credible evaluation of program fidelity within the Hoosier Sport intervention.

### 3.4. Measures

The Consolidated Framework for Implementation Research (CFIR) was used to guide the evaluation and selection of measures for the Hoosier Sport program. CFIR was developed as a broad, inclusive, and adaptable framework to understand the potential barriers and facilitators that a program has throughout the implementation process [28,29]. CFIR has been used by practitioners and researchers in developing and implementing physical activity programs for children [25]. To further evaluate the implementation, the StaRI checklist was used as guidance.

#### 3.4.1. Trial-Related Feasibility Indicators

**Recruitment Capability.** Recruitment of college students as implementers was measured as the number of eligible Hoosier Sport research team members that actively participated in attending the intervention at the middle school partner site compared to the total number of Hoosier Sport research team members. Child participant recruitment was measured as the number of 6th and 7th grade students who enrolled in the program compared to total school enrollment.

**Retention.** Retention of college students was measured through the proportion of enrolled participants who were actively engaged throughout the full intervention. This allowed understanding of the program’s ability to retain student implementers throughout its implementation and looked at the comparability between its recruitment and retention of facilitators. This metric also allowed for understanding the program’s ability for its members to cover other members who could not attend every implementation day due to various reasons (academics, personal, etc.). Retention for child participants was measured as the number of enrolled student participants who remained in the program throughout the 8-week time span.

**Barriers and Facilitators of Implementation.** This study used the barrier buster tool to understand team members’ beliefs and views of the Hoosier Sport program prior to implementation and then again 5 weeks after the program had been implemented. The barrier buster tool was developed by interviews with clinicians to create a simplified and easy-to-use version of CFIR that uses the 14 most important constructs, identified by the clinicians, from 4 out of the 5 domains [30]. The authors of this tool used an objective criterion to assess the pragmatism of a measurement instrument, i.e., barrier buster tool, and found the tool to be relatively pragmatic [31].

#### 3.4.2. Intervention-Related Feasibility Indicators

**Attendance.** Attendance was assessed for student implementers and child participants separately. Both were defined as the proportion of total intervention days attended compared to the total number of intervention days planned.

**Cost.** Cost was defined as the total amount of money used to conduct the intervention, excluding faculty personnel costs (i.e., percentage of salary coverage).

**Fidelity.** Fidelity, defined as assessing if the program was delivered as intended, was measured using two systematic observation tools, one for implementers (i.e., SOSPAN) and one for child participants (i.e., SOFIT). For implementers, we used the SOSPAN, which was developed to measure staff behaviors and management during school-related programs [32]. This tool informed us about how to better prepare program activities and understand ways to limit staff inhibiting behaviors while maximizing interactions with children and aiming to enhance child participation in physical activity [33,34]. For children, we used SOFIT to assess [1] the time spent in each lesson context and [2] the types of activities observed within each lesson context. SOFIT was developed in 1991 and has been used widely by children’s physical activity researchers [35,36].

### 3.5. Data Analysis

For our descriptive analysis, we calculated means and standard deviation for continuous variables and percentages for categorical variables. Since this was a pilot study with a primary aim of assessing feasibility, rather than hypothesis testing, data were primarily presented in a descriptive manner (e.g., SOSPAN, SOFIT, barrier buster tool) with the goal of continually refining and improving outcomes in subsequent iterations of the implementation strategy. The results of the barrier buster tool survey were examined as raw percentages and counts. Intervention-related feasibility indicators were analyzed against an a priori threshold of 80% or greater to determine success.

For the SOFIT analysis, we utilized percentages to break down the levels of physical activity for each relevant lesson context. This breakdown allowed us to understand the intensity and engagement levels of participants across various activities within the program. For the SOSPAN analysis, percentages were used to quantify the duration of key behaviors exhibited by staff and child participants during a 45 min lesson. Regarding other feasibility indicators, we evaluated trial-related and intervention-related aspects such as recruitment, retention, attendance, and cost. Specifically, we assessed the feasibility of college students serving as implementers at the remote rural site through their recruitment and retention rates. We examined attendance by calculating the percentage of participants present, providing insights into participant engagement and program utilization. Also, our cost analysis involved identifying the total expenses associated with implementing the program. For continuous variables used in statistical analysis, we assessed the normality of the data using the Shapiro–Wilk test prior to applying any statistical methods. Analysis was performed in R (version 4.0.3; R Core Team, Vienna, Austria), and the level of significance was set to alpha = 0.05.

## 4. Results

### 4.1. Trial-Related Feasibility Indicators

**Recruitment Capability.** There were 5 team members who actively participated in attending the intervention compared to a total of 15 team members. Thus, recruitment capability was 33%. One of the team members was the faculty advisor, three of the team members were graduate students, and one team member was an undergraduate student. We successfully recruited 23.3% of the total middle school enrolling 35 out of 150 students.

**Retention**. All five recruited team members who joined Hoosier Sport at the start of the program stayed throughout this study. Thus, retention was 100% for college student implementers. Of the 35 child participants recruited, we retained 33 participants, achieving a retention rate of 94.3%.

**Barriers and Facilitators of Implementation.** The barrier buster tool was completed by seven Hoosier Sport research team members during the pre-implementation of Hoosier Sport. The seven respondents answered all 14 questions, and there were no missing data. Of the 98 total item responses, 78.6% of the responses agreed with the statements, 15.3% of the responses were neutral, and 6.1% responses disagreed with the statements. Table 1 shows the results from each item of the barrier buster tool.

**Attendance.** There were at minimum two team members at each program day. The maximum number of team members was five on a given day. For the children, participants attended 80.2% of the sessions in the first 4-week segment focused on strength training, and 88.5% of the sessions in the second 4-week segment focused on basketball.

**Cost.** The cost to conduct the intervention was USD 12,050, including implementer transportation, sport equipment, accelerometers, graduate student financial support, and participant incentives.

**Fidelity.** Fidelity was assessed with two synergistic tools assessing if the program was delivered as intended: SOSPAN (assessing implementer’s behaviors) and SOFIT (assessing children’s PA). For SOSPAN, there were a total of 80 scans completed by the primary and reliability scanners. All individual scans were found to be >80% matching between the primary scanner and reliability scanner. Figure 2 shows the SOSPAN results. A total of 100% of scans found that staff never withheld physical activity or used physical activity as a form of punishment.

### 4.2. SOFIT Results During Basketball

The distribution of student activities within the basketball context, as illustrated in Figure 3, showed significant variations across different lesson contexts. Management contexts predominantly involved standing activities, supplemented by occasional instances of walking or moderate activities for tactical purposes. In game-related contexts, students were frequently engaged in vigorous activities, interspersed with occasional periods of moderate activity. In terms of fitness activities, students primarily participated in moderate levels of physical activity.

### 4.3. SOFIT Results During Strength Training

In the strength training segment, as depicted in Figure 4, student activities displayed significant variability across various lesson contexts. During management scenarios, students were standing. Skill-focused activities were predominantly characterized by vigorous activity levels. Similarly, games and fitness contexts primarily featured vigorous activities. Fitness sessions occasionally involved walking or moderate activities if students were not actively engaged in vigorous exercises.

## 5. Discussion

The objective of the prospective longitudinal study was to evaluate the implementation of Hoosier Sport in a rural middle school using a multi-component evaluation approach. There were four main findings from this study. First, for feasibility indicators, college students were successfully recruited, retained, and attended the intervention above a priori thresholds. Second, the SOSPAN and SOFIT tools identified positive fidelity for program delivery for both the implementers and the child participants. Third, the findings from the barrier buster tool found that team members generally agreed with all statements, pointing to promising system-level acceptability. Fourth, the findings from this study highlight the importance and potential value for continued refinement of early-stage pilot testing and iterative implementation evaluation in line with the ORBIT model. Furthermore, the four main findings confirmed the hypothesis that Hoosier Sport implementation was feasible, delivered as intended, and identified refinement opportunities for future intervention testing.

First, Hoosier Sport proved feasible for service-learning/college students to serve as implementers, with team members successfully recruited, retained throughout the study period, and meeting their intended attendance targets. College students had autonomy in the scheduling days they were available to attend the program, but there was an expectation of recurring attendance. When implementers had to miss a day due to a prior commitment or personal emergency, the rest of the team was able to cover for them, ensuring that at least two facilitators were present on every program day. These measures served as critical indicators of the overall implementation strategy—designed to connect the pipeline of college students to communities. Our university alone has approximately 40,000 college students in any given year, and these students need community-based and practical experience, as provided in the present study and research trajectory. While this approach for cultivating campus–community partnerships and using trained college student mentors as implementers sounds promising, there is a need to build evidence of its efficacy.

A key aspect of this study was its focus on under-resourced rural populations, which are often overlooked in research or included only at the start of clinical trials, missing out on later advancements. Rural populations face unique sociocultural and transportation barriers, making them harder to reach compared to urban populations closer to research institutions. The partner school was located approximately 50 miles from the university campus, meaning the implementation team had a total round-trip commute time of approximately 2 h each intervention day. This additional barrier, compared to urban or more local school district partnerships, points to the notable nature of the successful intervention-related feasibility indicators. Furthermore, while rarely reported, cost is one of the most important variables to consider when designing and delivering health promotion interventions. The cost of the present study was USD 12,050. The greatest expense in most programs was personnel, but this study benefited from the university’s curriculum, which allowed students to earn 1–3 credits per semester for community-based research. Many students, especially first-generation students who lack parents or mentors who attended higher education institutions are unaware of the need to gain extracurricular research lab experience if they hope to pursue graduate degrees [37,38]. Hoosier Sport has benefited from the general lack of university programs that offer community-based research experiences. Many programs investigate isolated research projects, such as traditional exercise interventions, but Hoosier Sport is attempting to build a larger system integrating the college student pipeline and university curricular programming to under-resourced communities. The college student workforce/pipeline is an essential part of this model. One of the direct benefits compared to a standard programming model is the cost savings by explicitly providing curricular credit and/or implicit need for students to gain practical/research experiences if they hope to have a competitive graduate school/job application.

The second key finding was that there was high fidelity in implementation delivery for both the college student implementers (based on SOSPAN) and the child participants (based on SOFIT). One aspect of why we believe these fidelity findings to be so positive is that we used a strong theoretical framework (i.e., psychological needs satisfaction) when designing this study [37]. The implementers also regularly discussed feedback on each session during the transportation time to and from the school site, and this feedback existed within the lens of trying to satisfy autonomy, competence, and relatedness within the implementation strategy. In line with research emphasizing the importance of psychological needs to well-being and physical activity, [39] when children’s psychological needs are met during physical activity, it is more likely that positive SOSPAN results will be observed. Previous research using SOSPAN has shown that children are more active when staff promote and participate in physical activity [33,40]. Some examples of psychological need satisfaction in action include but are not limited to the following: staff verbally encouraging during physical activity programs and ensuring that physical activity was never used as a form of punishment or taken away from children for behavioral reasons (i.e., autonomy); elimination games were almost never used during sessions (i.e., competence); and small teams were frequently used which encourages greater inclusivity and teamwork (i.e., relatedness). These physical activity strategies (e.g., small teams/groups, lack of elimination games) have been supported in other studies as effective for helping kids to participate in more physical activity and spend less time standing around [41,42]. Importantly, in our previous work reporting intervention outcomes [23], the present intervention was successful in increasing the satisfaction of basic psychological needs in the child participants. Finally, the SOSPAN tool found that the implementers were involved in physical activity with the child participants about half of the time (i.e., relatedness). The shared identity of being a college student contributed to relatedness among the implementers, which, in turn, allowed for cohesion between team members. This cohesion and relatedness allowed the implementers to effectively promote and participate in the physical activity during the program. Further research should be conducted to have a deeper understanding of the relationship between staff engagement and promotion of physical activity and individual child physical activity levels. SOSPAN should continue to be used to increase feasibility in the Hoosier Sport intervention and future youth-based sports programming.

Additionally, the SOFIT assessment was a preliminary indicator for high fidelity in child-focused outcomes, evidenced by high physical activity participation rates and elevated physical activity levels among middle school students during the 45 min physical education class. The data showed that students were engaged in moderate-to-vigorous activity for most of the class, whether in basketball or strength training. Strength training sessions allowed more skill acquisition opportunities, highlighting a structured approach with vigorous activity followed by necessary rest periods. In contrast, basketball sessions involved sustained physical activity with minimal breaks, emphasizing continuous movement and endurance. These insights suggest that particular sports may focus on different lesson contexts and types of activities while still striving for and achieving high physical activity participation. Further, this built-in/intentional sport sampling is conducive to autonomy, competence, and relatedness in that children are provided with sport/exercise choices, different sports tailoring to different abilities and body types, and relatedness in that different sports will elicit different levels of engagement in different children. Tailoring training strategies based on these observations can improve skill development, physical fitness, and overall performance. Importantly, a key strength of the present study was its detailed reporting of lesson contexts. A prior systematic review [36] found that 75% of the studies included failed to provide this information. By including lesson context data, our study offers valuable insights into intervention design for promoting physical activity. Moreover, the present intervention significantly promoted MVPA, achieving rates well over 50% of the lesson period (95% for basketball and 94% for strength training). This far exceeds other studies, which report consistent MVPA ranging from 13% to 49.9% of the observed lesson periods [23,39,42,43,44].

Third, the barrier buster tool data identified two important barriers to implementation. The first barrier was that not all members of Hoosier Sport had open lines of communication. This is due in part to Hoosier Sport being a relatively new pilot program as it launched in the spring of 2023. It has been shown in other research that constant communication by the individuals implementing initiatives is needed to effectively discuss programming and activities [45]. The barrier buster tool is primarily used for practitioners, but other literature supports that a lack of communication between patients and practitioners led to a lack of awareness and education [46]. As Hoosier Sport continues to grow and refine its infrastructure, this finding highlights the importance of keeping communication systems at the forefront of development. The second perceived barrier identified by the barrier buster tool was that not everyone agreed that change was needed at the rural middle school. This finding speaks to the importance of language and taking a critical approach to community-engaged research. The survey item referred to whether the current situation at our middle school partner was “intolerable.” That word is inherently judgmental, so it is not surprising that not all respondents agreed with the statement. Hoosier Sport takes a critical approach to community-engaged intervention and implementation delivery, meaning we attempt to avoid taking a savior mindset when entering communities. By entering communities non-judgmentally, we attempt to build rapport and respect the strengths, resources, and cultures of communities. Our intervention and implementation process began with rapport building and needed assessment development as previous literature has identified that need assessments are important to understand the type of change warranted [45,46]. The findings from the barrier buster tool helped to complete our secondary objective of identifying constructs within 4 out of the 5 domains that can inhibit or promote successful implementation of Hoosier Sport.

This paper has identified several recommendations for the Hoosier Sport program to continuously refine its implementation. First, constant communication is needed between team members to identify barriers and facilitators of the program. Second, systematic observational tools are vital in the planning, facilitation, and evaluation of fidelity of sport-based youth development programming. Lastly, the continued development and commitment to the community-based partnership are crucial to the success of this program, ensuring it continues to benefit both college student mentors and middle school students alike. There were also several strengths identified in this study. First, the partnership between an academic institution and a local middle school allowed for a mutually beneficial relationship where college students can develop practical skills and middle school children can participate in a sport-based program to increase physical activity levels and be exposed to positive role modeling. Second, SOSPAN and SOFIT effectively promoted planning activities and implementation refinement that allowed for continuous movement by participants and discussion around the types of positive reinforcement behaviors that encourage children to want to participate in the activities. Third, there is potential for the Hoosier Sport implementation strategy to continue refining its implementation methods in subsequent testing. Further, this college student driven implementation model holds many lessons learned that could be utilized by other academic institutions to connect with local communities.

There were several limitations identified in this study. First, the present study was a pilot study with a small sample size. Correlation cannot be inferred from the results of this study as there was no control group (e.g., comparator school). The number of data points and follow-up period were limited. Further, the SOSPAN scanning tool is limited in its ability to produce data that can be generalized. While there were 40 scans, the scanning tool only allows for staff and child participants to be scanned for one second each scan. Therefore, it can be assumed that the observation collected during that one second does not encapsulate the behaviors and activity level of the program overall. Lastly, implementers following the pre-planned lesson plan schedule set by the implementation team was not always in the best interest of the child participants. Certain activities and programming had to be adapted to the needs, wants, and fluctuating energy levels and interests of the child participants. For example, activities originally designed to encourage vigorous physical activity were sometimes adjusted to accommodate lower energy levels or to better align with the children’s preferences, such as incorporating more game-based approaches. Team members faced challenges such as maintaining engagement during longer sessions, ensuring inclusivity for children with varying skill levels, and balancing structured activities with flexibility to respond to participants’ feedback. These challenges were addressed through ongoing observation, team debriefs, and iterative modifications to the activities, ensuring the program remained enjoyable while encouraging consistent participation. As Hoosier Sport is early in its implementation, trial and error was a factor as team members attempted to develop activities that were enjoyable while encouraging high levels of physical activity participation for all children.

## 6. Conclusions

Hoosier Sport’s use of multi-component evaluation tools, such as SOSPAN and SOFIT, underscores its commitment to high-quality implementation and continuous improvement. By ensuring high fidelity in program delivery, the program exceeded a priori progression thresholds for implementation and physical activity engagement. This high level of physical activity engagement is essential for public health, addressing critical issues like childhood obesity and cardiovascular health, while promoting habits that encourage lifelong fitness. Moreover, the program’s reliance on college students as implementers highlights a sustainable model that benefits both the students and the middle school participants. College students gain valuable practical experience and academic credit, while middle school students receive mentorship and role modeling from young adults. This symbiotic relationship enhances the educational experience for college students and enriches the developmental environment for middle school children. This symbiotic relationship enhances the educational experience for college students and enriches the developmental environment for middle school children. However, the program also faced notable challenges, including fluctuations in child participant energy levels and engagement, as well as logistical difficulties in aligning program activities with the school schedule and managing resource limitations in a rural setting. Addressing these challenges required iterative modifications to activities and close communication with school staff. Despite these obstacles, Hoosier Sport exemplifies a model that other programs might replicate to foster both personal development and community health improvement, setting a benchmark for future physical activity initiatives.

## Figures and Tables

**Figure 1 ijerph-21-01645-f001:**
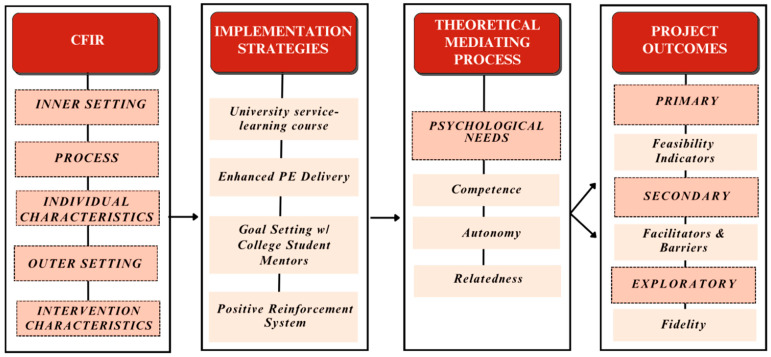
Hoosier Sport conceptual model.

**Figure 2 ijerph-21-01645-f002:**
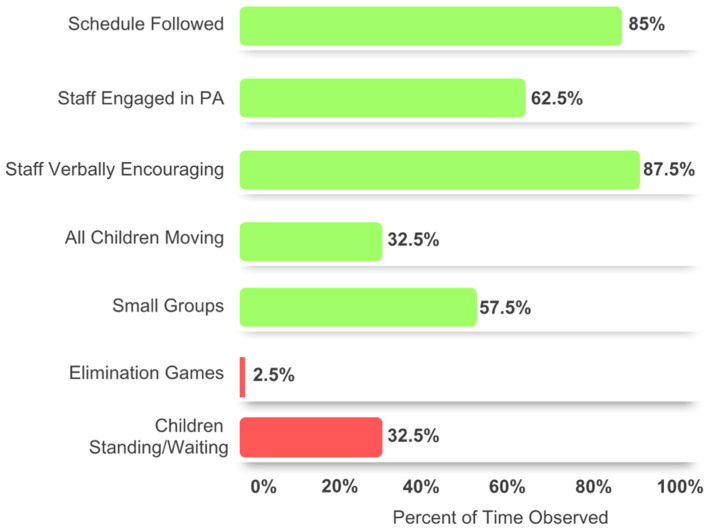
SOSPAN observations in a 45-minute class period. Each bar represents a key behavior measured, while the percentages reflect the frequency of those behaviors during a single class period. Green is for positive behaviors that we want our program to encourage, while red is negative behaviors that we do not want to minimize.

**Figure 3 ijerph-21-01645-f003:**
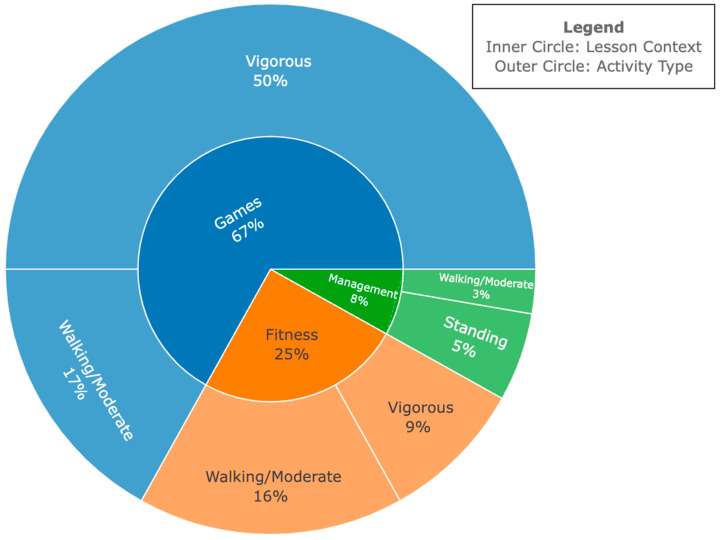
SOFIT results for the basketball segment. The chart illustrates two key aspects of lesson time during the basketball segment. The inner circle shows the time spent in each lesson context, including the percentage of total lesson time (e.g., games accounted for 67% of the time in a 45 min lesson). The outer circle details the types of activities observed within each lesson context, indicating the percentage of time spent on each activity (e.g., walking/moderate, vigorous) for lesson contexts.

**Figure 4 ijerph-21-01645-f004:**
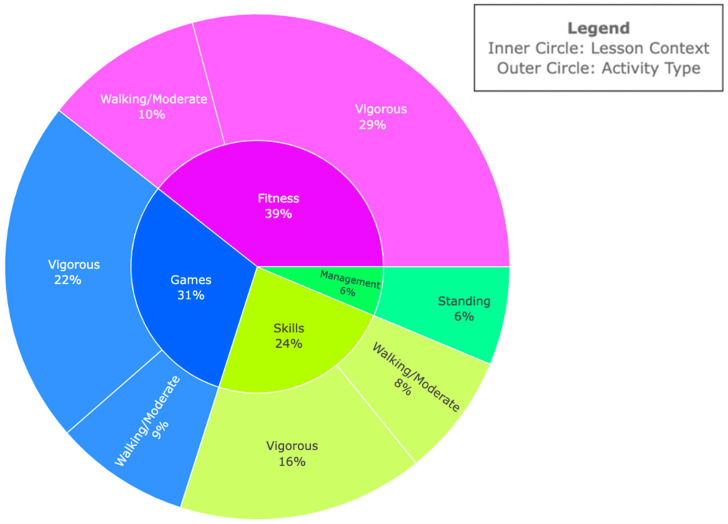
SOFIT results for strength training segment: the chart illustrates two key aspects of lesson time during the strength training segment. The inner circle shows the time spent in each lesson context, including the percentage of total lesson time (e.g., games accounted for 31% of time in a 45 min lesson). The outer circle details the types of activities observed within each lesson context, indicating the percentage of time spent on each activity (e.g., walking/moderate, vigorous) for lesson contexts.

**Table 1 ijerph-21-01645-t001:** Intervention-related feasibility indicators.

Statement	Time	Agreement Level (% Agreed)	Perceived Effect (% Strong)
**1. Hoosier Sport team members regularly seek to understand the needs of WRV children and make changes to better meet those needs.**	Pre-implementation	100%	100%
	Mid-implementation	100%	100%
**2. I have open lines of communication with everyone needed to improve the lives of the WRV community.**	Pre-implementation	42.9%	71.4%
	Mid-implementation	71.4%	85.7%
**3. I have access to data to help track changes in outcomes.**	Pre-implementation	71.4%	85.7%
	Mid-implementation	85.7%	71.4%
**4. The efforts of Hoosier Sport are aligned with project goals.**	Pre-implementation	100%	100%
	Mid-implementation	85.7%	85.7%
**5. The Hoosier Sport mission is aligned with the values of WRV community.**	Pre-implementation	85.7%	100%
	Mid-implementation	85.7%	85.7%
**6. The Hoosier Sport program is compatible with existing WRV policies and procedures.**	Pre-implementation	85.7%	100%
	Mid-implementation	42.9%	71.4%
**7. The structures and policies in place at WRV are conducive to the Hoosier Sport Mission.**	Pre-implementation	57.1%	85.7%
	Mid-implementation	71.4%	85.7%
**8. We have sufficient space at WRV to accommodate the Hoosier Sport Program.**	Pre-implementation	85.7%	71.4%
	Mid-implementation	42.9%	85.7%
**9. We have sufficient time dedicated to the program to be effective.**	Pre-implementation	85.7%	100%
	Mid-implementation	85.7%	85.7%
**10. We have other needed resources to make the change (staff, money, supplies, etc.).**	Pre-implementation	71.4%	85.7%
	Mid-implementation	85.7%	85.7%
**11. Team members in Hoosier Sport see the current situation at WRV as intolerable and that change is needed.**	Pre-implementation	14.3%	14.3%
	Mid-implementation	42.9%	42.9%
**12. People here see the advantage of implementing Hoosier Sport versus an alternative change.**	Pre-implementation	100%	100%
	Mid-implementation	71.4%	57.1%
**13. Higher level leaders are committed, involved, and accountable for the planned improvement.**	Pre-implementation	100%	100%
	Mid-implementation	85.7%	85.7%
**14. Leaders I work with most closely are committed, involved, and accountable for the planned improvement.**	Pre-implementation	100%	100%
	Mid-implementation	85.7%	85.7%

Notes: *n* = 7 participants; pre-implementation: before program began; mid-implementation: 5 weeks after program began. These items were adapted from The Barrier Buster Tool [30].

## Data Availability

The datasets used and/or analyzed during the current study are available from the corresponding author upon reasonable request.

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
