# Peer review of "Evaluation of a Pilot College Student-Based Physical Activity Implementation Strategy in a Rural Middle School"

_ijerph, 2024, doi:10.3390/ijerph21121645_

Round 1
Reviewer 1 Report
Comments and Suggestions for Authors
The authors must be commended for carrying out a study regarding the college student-based physical activity implementation strategy in a rural middle school. This topic is interesting and very important. The research methodology used in the study is appropriate, and the manuscript is written with great clarity. However, some issues need to be taken into consideration. Please find my comments in the following text.
Title
Since you emphasised in the text that this is a pilot study, I strongly suggest also emphasising it in the title.
Abstract
Please a statistical analysis used in the study.
Introduction
Please explain the structure of the Hoosier Sports physical activities in more detail.
Methods
Sample: Why did you include this specific number of participants? Please elaborate.
Child participants: Please add an ethical statement; e.g. this study was conducted in accordance with the Declaration of Helsinki...
Please explain the structure of the implemented sports activities in more detail.
Line 266: Please add manufacturer information for the R.
Results
Table 1: I strongly suggest replacing a picture with a Table.
Discussion
Congrats on the discussion section.
Author Response
Please see the attachment in the box.

Reviewer 2 Report
Comments and Suggestions for Authors
Introduction
The importance of the topic could be better supported with additional examples. In particular, providing more details on the challenges of accessing physical activity in rural schools would help the reader better understand the situation.
Further information about how the Hoosier Sport program differs from similar programs would strengthen its originality. For example, comparisons with other programs could be made.
It would be helpful to provide specific examples of how the study addresses the existing knowledge gap in the field.
The problem statement and hypothesis could be more clearly articulated.
Method
The method section is generally well-structured. However, I believe adding more details and clarification in the following areas would highlight the strengths of the research:
Suggestions
- More detailed information on the participant selection process and demographic data.
- Additional information about the effectiveness of the training process for the implementers.
- More details on the validity and reliability of the measurement tools.
Discussion
While it is mentioned that the program needs to be adapted to the participants' needs, more information on the challenges encountered in the study and how these challenges were overcome would be valuable, especially for future research.
Suggestions for the Conclusion
The collaboration between an academic institution and a local school has created a mutually beneficial relationship for both parties. This partnership has provided opportunities for students to develop practical skills and for children to engage with positive role models. Specifically, the suggestions section could benefit from including recommendations on how such collaborations can be sustained in the long term.
Author Response
Please see the attachment in the box.

Reviewer 3 Report
Comments and Suggestions for Authors
Dear authır(s),
Line 120, 123: "aim" should be used instead of "objective".
It should be stated which gap in the literature the study will fill. The aim should be written better. The problem statement is well defined, but the original value should be written better.
n: Is 7 intervention sufficient? Was a power analysis or other measurement used?
Table 1 should be in picture format, word format.
Discussion: The introduction is too long. Line 334-376. Descriptive information is represented. Discussion should be simplified. The results obtained should be compared with the literature.
The findings were added to the discussion
Line 460-485: Too long, repeating the same things.
Method: adequate
Findings: Adequate
References: Proficient.
Author Response
Please see the attachment in the box.

Reviewer 4 Report
Comments and Suggestions for Authors
Dear authors,
Below is feedback and I have also included comments and suggested changes in the attached PDF.
This is one of the best papers that I have reviewed to-date. Below is feedback and I have also included comments and suggested changes in the attached PDF.
Introduction
The introduction is clear explanation of the issues that the study then intends to overcome. Other researcher is appropriately cited as needed. Your research appears to be quite simple, but it is definitely innovative and aligns with research in other contexts for learning such as drug and online safety education that shows older peers delivering aspects of the teaching having a significant effect.
In the abstract and also in the introduction, you refer to the "power of sport". I feel that the term "power" is very broad and does not convery the specific meaning which appears to actually be "context". Therefore you might be best to refer to "context of sport".
Method
The methodology is of the highest quality with fidelity checks of the implementation of the research study. No faults or issues here.
Results
Clear and self-explanatory. I do ask to create Tables with the data rather than pasting in images.
Discussion
Aligns the results with an indepth sythesise of the other research.
Conclusion
This section could be improved.
You mention that the program impacts childhood obesity and cardiovascular health. Emphasising the importance of these outcomes could reinforce the program’s public health relevance. I suggest “This high level of physical activity engagement is essential for public health, addressing critical issues like childhood obesity and cardiovascular health, while promoting habits that encourage lifelong fitness.”
Consider ending with a sentence that underscores the program’s broader implications for similar initiatives or future research, which would provide a forward-looking perspective. My suggestion is “Hoosier Sport exemplifies a model that other programs might replicate to foster both personal development and community health improvement, setting a benchmark for future physical activity initiatives.”
Kindest regards

Author Response
Please see the attachment in the box.

Reviewer 5 Report
Comments and Suggestions for Authors
1. The recommendation is to include the paper's purpose in the abstract.
2. Congratulations to the authors for introducing the Contributions to the Literature part, with all that it contains.
3. Please include the sample size calculation and/or statistical power. In the worst case, it is advisable to specify this in the form of study limitations.
4. The recommendation is to put more emphasis on the inclusion and exclusion criteria part.
5. In the discussion part, after the objectives paragraph, please kindly specify whether the study's hypothesis has been confirmed.
6. Importantly, in our previous work reporting intervention outcomes (40)… Please note that the bibliographic source number 40 does not have your corresponding bibliographic article, so please reword or insert the corresponding source.
7. A recent systematic review (36) found that 75% of the studies included failed to provide this information. That systematic review you refer to is not very recent, so please rephrase.
8. There are some outdated bibliographical references, so please replace the sources with numbers 3, 10, 19, 24, 28, 35 and 38.
Comments on the Quality of English LanguageModerate editing of the English language is needed.
Author Response
Please see the attachment in the box.

Reviewer 6 Report
Comments and Suggestions for Authors
Well-structured research and with adequate statistical analysis, this is an interesting topic for healthy development and coexistence among school children.
However, I believe the following improvements could be made in the paper:
In the introduction, previous studies evaluating the effects of physical activity programs in similar communities should be included, if not found, at least the effects of programs in the school population.
In the methods, it is necessary to describe the conditions and schedule under which the evaluations, questionnaires, and interviews were conducted. It is necessary to explain the place and group or individual application of the whole procedure.
In case of statistical analysis, it is necessary to point out the applied normality test.
In the discussions, it is necessary to discuss more extensively the results related to the conformation of small work groups in relationships and inclusion among students, as well as the use of non-competitive game. These seem to be important points in the program used.
In the conclusions, it is necessary to include the specific benefits found, both for university students and schoolchildren, as well as to point out the main difficulties detected.
Author Response
Please see the attachment in the box.

Round 2
Reviewer 5 Report
Comments and Suggestions for Authors
Congratulations on your hard work in producing this scientific material!